# Integrating Societal Issues with Mathematical Modelling in Pre-Service Teacher Education

Lisa Steffensen * and Georgia Kasari

Department of Language, Literature, Mathematics and Interpreting, Faculty of Education, Arts and Sports, Western Norway University of Applied Sciences, 5020 Bergen, Norway; georgia.kassari@hvl.no
* Correspondence: lste@hvl.no

**Abstract:** The complex societal phenomena occurring in our daily lives and the ongoing curricula demands of mathematics education imply the responsibility of teachers to discuss societal issues with their students in mathematics classrooms. Yet, the ways in which teachers respond to these demands are neither given nor straightforward. In this case study, we aim to understand how pre-service teachers are introduced to addressing societal issues during mathematical modelling activities through the examples utilised by a teacher educator. Theoretical perspectives from socio-critical modelling are used to investigate examples from a mathematics teacher education course where socio-critical perspectives of modelling activities were addressed. We found that the teacher educator included multiple activities with contexts relevant to pre-service teachers, such as littering, body images, and oil spills, and focused on problem posing. Also, the complexity of socio-critical modelling activities was illustrated by bringing various perspectives and alternatives, and a need for commitment to action and assuming responsibility was discussed. Our findings conclude that mathematical modelling can be one way of incorporating socio-critical issues in teacher education to prepare pre-service teachers to be, and become, critical and responsible citizens, yet, doing so requires the engagement of a community of teacher educators.

**Keywords:** mathematical modelling; socio-critical perspectives; mathematics teacher educator; teacher education

## 1. Introduction

Society and citizens face numerous challenges, such as climate change issues, sustainability, pandemics, and refugee crises. The challenges are often multifaceted and require approaches that take into consideration political, economic, or ethical dimensions. To prepare students to deal with these challenges as sensitised and responsible citizens, educators, including teachers, pre-service teachers (PTs), and mathematics teacher educators (MTEs), also need to become sensitised and assume responsibility for their role in addressing complex socio–political issues in their respective fields of practice.

In mathematics education, approaching real-world problems often occurs through engaging in modelling. Mathematical modelling refers to the process of representing real-world phenomena or systems and translating them into a mathematical framework to analyse, predict, and understand a real-world problem. Previous studies have indicated that mathematical modelling competencies are central to educating responsible citizens [1,2] because they require making mathematical and non-mathematical decisions [3]. However, according to [4], modelling activities around challenging issues often focus on transforming real-world situations into mathematically solvable problems rather than addressing the complexities of the issues themselves or reflecting on the use of mathematical models. To avoid modelling activities becoming just a context to "dress up" mathematical problems, there is a need to move beyond mathematising problems and building students' experiences relevant to real-world situations, such as taking political action. Engaging students to use mathematical modelling as a tool to investigate issues of the world and society, challenge

inequities, and take action has been discussed in previous research, such as [5–11]. In particular, ref. [11] highlighted that modelling activities overlap with mathematics for social justice by dealing with controversial real-world situations, reflecting on alternative solutions, and supporting students as "competent knowers and doers of mathematics" (p. 1).

Mathematical modelling is included in an increasing number of curricula world-wide [12], such as in Germany [13] and Norway. In Norway, where our study takes place, modelling recently became one of six core elements in the mathematics curricula reform for grades 1–10 [14]. According to the reform, students should learn about how models are used and become able to critically evaluate modelling in society. For instance, a competence aim for fourth-grade students entails learning to "model situations from one's own everyday life and explain one's own thought processes" (p. 8).

However, it can be challenging for teachers and students to respond to curriculum demands to do with mathematical modelling [15]. When there are additional expectations to address socio–political issues, more pedagogical challenges may emerge as teaching becomes more uncertain. Therefore, the role of teacher education in preparing future teachers to deal with the uncertainties and various demands of combining mathematical modelling activities with knowledge about society needs to be further examined. PTs need to be supported to engage in modelling activities regarding societal issues with their students [16,17]. However, little is empirically known about how MTEs support them during mathematics teacher education courses.

In this paper, we discuss a case study to understand how mathematical modelling and socio-critical issues can be integrated into teacher education. Based on the case study of one MTE, our focus is on understanding how mathematics teacher educators exemplify associations of modelling activities with societal issues. To investigate this, we use data from an MTE's planning and implementing of three introductory workshops on mathematical modelling to second-year PTs of grades 1–7 as part of a five-year teacher education program in Norway. Understanding how MTEs use examples of integrating societal issues in mathematical modelling activities can provide insights on supporting PTs to embrace the uncertainties and engage in those issues with their students.

## 2. Integrating Mathematical Modelling and Societal Issues in Teacher Education

Societal issues within mathematics education can involve teachers' and students' explorations of how mathematics intersects with broader social, economic, and political contexts. In particular, perspectives from critical mathematics education (CME) have focused on social justice issues, questioning the role and influence of mathematics in society, and how power dynamics and social inequalities can be reinforced or challenged through mathematical practice [18–21]. There are many forms of social justice and defining it is not straightforward, because what is just to one person might be unjust to someone else [22]. Social justice issues can include, for example, the fair distribution of wealth, resources, opportunities, and privileges or inequalities in society (Oxford Learner's Dictionaries [23]). According to [21], learning about issues of justice and injustice in mathematics school classrooms is not a matter of telling students what these issues are, but, rather, engaging them in developing experiences about what social justice is, and how it is formed. Teaching and learning mathematics with this perspective aims to support students in expressing what they consider as social and economic inequalities, oppression, and exclusion or address structural discrimination and power relations benefitting certain perspectives (e.g., valuing rapid economic growth at the expense of environmental problems and climate change). Although CME initially focussed on social justice problems, ref. [21] introduced the term environmental justice to include environmental issues as part of the concerns of CME. For example, ref. [24] analysed students' involvement in a modelling task to estimate the state of the Great Barrier Reef by 2050. They described that the students became aware of environmental injustices and their influence, not only on the lives of the Australasian people but also on their own. Engaging in this activity allowed students to combine their

understanding of society with mathematics and develop a sense of agency and political change [24]. Although conducting and defining social justice is a complex and ongoing endeavour, rather than a single product, the aims of developing a social justice stance are not separate from mathematics learning goals, according to [25]. Ref. [25] discussed three specific mathematics learning goals (reading the world with mathematics, mathematical empowerment, and changing dispositions towards mathematics) as aligning with three goals of learning about social justice (socio–political consciousness, agency, and positive social and cultural identities). In this paper, we have a broad understanding of social justice, including societal issues, such as environmental justice, marginalisation, stereotypes, biases, power structures, and cultural and linguistic diversity.

Several research studies have investigated mathematical modelling within teacher education. Ref. [26] discussed experiences as MTEs from modelling courses and mentioned challenges they needed to deal with, such as balance theory and practice, appropriate teaching strategies, and content knowledge (e.g., modelling cycle, goals/perspectives, types of tasks, solving and creating modelling tasks, and planning and practising lessons). Ref. [16] described four dimensions of pedagogical content knowledge for teachers: a theoretical dimension, a task dimension, an instructional dimension, and a diagnostic dimension. Ref. [27] found that PTs could develop their competencies in these four dimensions when their experiences as learners of modelling were combined with their experiences as teachers of modelling. For example, PTs should develop competencies such as choosing appropriate modelling tasks and taking various approaches. Related to the findings of [27,28], the authors found that when PTs were provided with rich modelling tasks (e.g., good-quality tasks where PTs engaged in real-life problems where they made choices, developed representations, and engaged in decision making) and an emphasis on modelling, they demonstrated an understanding of mathematical modelling.

There are also some studies within teacher education focusing on socio-critical modelling perspectives. Ref. [10] explored how PTs connected mathematical modelling and social justice issues by investigating what kind of problems they posed. They found that the PTs posed problems of social justice issues involving both micro-level (e.g., individual or community issues) and macro-level (e.g., broader issues involving political and economic structures involving disadvantages for some groups). From the perspective of problem posers, ref. [10] proposed a conceptual framework for social justice-oriented mathematical modelling tasks consisting of social justice issues, realistic context, model development, and shareable processes. According to PTs' reflections in the study of [29], some problems were "too simplistic" for students, while other students had little background in posing open questions. Nevertheless, in her study, PTs were able to develop their open problem posing after observing their peers' problems and designing multiple variations of a potential task inspired by authentic pictures they took from their environment.

Ref. [30] integrated a critical mathematics education perspective with a socio-critical approach to modelling, encouraging PTs to reflect critically on the application of mathematical models in society. When the PTs developed modelling activities for students, ref. [30] found that community interactions and classical and critical knowledge were needed for developing authentic modelling tasks and fostering good modelling experiences for students. Ref. [9] suggested combining mathematical modelling with culturally responsive, social justice-oriented mathematics. In their study, an ongoing water crisis was used as an example to explore social and environmental justice issues. Their findings suggested that engaging in tasks addressing these issues increased PTs' and teachers' awareness of systemic injustice while strengthening their mathematical competencies. Refs. [31,32] described how MTEs stimulated critical mathematical discussions on the use of indices, such as the body mass index (BMI), inspired by modelling as critiqued by [7]. The primary school teachers in their course were shown a picture of a muscular athlete with a BMI of 35.8 kg/m$^2$ (which would put him in the obesity range). In indices such as BMI, mathematics is often implicit; still, it is crucial in defining the phenomena. Facilitating

critical discussions amongst students could be one way of critically reflecting on the role of mathematical models in society.

Ref. [33] described mathematical modelling as one way of addressing socio–political issues related to injustice. He used the context of the Charlottesville rally (far-right groups' protests) with his PTs to explore issues of segregation in U.S. schools and racism in society. Drawing on research literature and his own experiences, he proposed a framework highlighting the importance of building a socio–politically oriented community. The knowledge about socio–political issues, curriculum, and mathematical teaching inform how to mathematise socio–political problems. He further highlighted that a key to building a socio–politically oriented community is having a network of colleagues committed to exploring societal issues, as well as resources (e.g., social media sites on equity and mathematics education). Ref. [34] described a modelling activity conducted by one group of PTs in their practicum. The PTs utilised pictures of refugees to encourage students to pose questions about the refugee crisis and supported them in posing examinable mathematical questions.

The choice of context can reflect the educational purposes of modelling as a purposeful activity. Ref. [35] exemplified contexts such as environmental problems, migration, and inequality, yet mentioned that identifying contexts that would work in classrooms worldwide is difficult. This is because different cultural experiences among students and teachers can have an impact on how one perceives things and lead to numerous variations of models. Related to this, ref. [16] explained that the aim and justification of modelling influence the choice of context or examples. For instance, a cultural justification, where relations to the extra-mathematical world are involved, authentic, real-world examples showing the role of mathematics more explicitly are suggested contexts. For example, ref. [36] discussed a modelling activity about ocean trash, where seventh-grade students investigated the Great Pacific garbage patch. This activity allowed students to engage both with mathematical thinking and knowledge about society since they posed questions about the origin of the trash, its effects on wildlife, and what actions could be taken in addition to designing a model to convey the size and density of the garbage patch and discussing various approaches.

Several examples of modelling contexts scale up or down items from real life, e.g., the Giant shoe problem and the Oldenburg's pick-axe with the Kassel Hercules [16,37,38]. Another variant of such proportional thinking is the Barbie doll. Ref. [39] asked students what Barbie would look like if she were the height of an average woman as an example of proportional reasoning. The differences between the doll and humans led them to discuss body images and eating disorders. Exposure to images of thin, idealised bodies, which dominate the social media, can contribute to body dissatisfaction. Ref. [40] described that Barbie has unrealistic measures and "promote(s) harmful weight attitudes including thin-ideal internalisation in young girls" and can "negatively influence children's feelings about their own bodies and their eating behaviours". Ref. [41] let her PTs read the article by [39] in a course about social justice in mathematics education. Her PTs' reactions to the paper were that it was "too feminist" and did not match their expectations of learning practical "tips and techniques" about how to integrate social justice issues in the mathematics classroom (p. 210). However, ref. [41] said the activity "represented tangible ways to both understand the marriage of mathematics and social justice and to feel good about doing what they could to address social justice issues in/through mathematics through more meaningful 'real-life' connections" (p. 210). Ref. [2] highlighted how the context of the COVID-19 pandemic has demonstrated that citizens need to understand how mathematical models function, help us to understand, predict, and overcome crises. It has also shown that citizens and decision-makers should learn to critically evaluate reports affecting how we act during a crisis and deal with the inherent uncertainty. They argued that these new demands had expanded the learning of modelling in education to not only include learning to apply mathematics to real-world contexts but also to include learning decision making, dealing with uncertainty, and adopting critical thinking. Issues of inequality were used as context when ref. [42] described a modelling task involving the distribution of wealth in the

world as a context "to unveil social and political phenomena, and promote informed critical position" (p. 6). This summarises in many ways how context is particularly important in socio-critical modelling activities.

Several studies involving societal issues and mathematics have reported challenges. Ref. [9] described reluctance to mathematise controversial issues, such as racial profiling or discrimination. However, they highlighted that an increasing number of MTEs invited PTs to investigate fairness, environmental issues, and economic justice. Ref. [33] explained that colleagues, leaders, or others could question the use of socio–political context in mathematics courses from views about what mathematics courses should entail. He underlined that PTs might feel overwhelmed by the time it takes to develop relevant socio–political tasks, experience pushbacks from colleagues, or uncertainty to navigate challenging conversations. He suggested that PTs could find support elsewhere, such as in like-minded colleagues, various resources, starting with less controversial topics (environmental issues), and curricula stating the relevance of real-world problems or mathematical modelling where the context is essential. Ref. [8] reported that although PTs stated that they would include real-world situations, they were ambivalent about having controversial topics and issues of injustice in mathematics education. Ref. [43] described it as time-consuming to find contexts that work well with both developing responsible citizens and competencies in modelling. Some teachers in their courses stated it was difficult to identify and address the mathematics involved in the context (e.g., they considered the context of plastic waste to be more connected to biology). Others stated it took more time to understand the context than the benefits from the mathematical outcome. Ref. [43] highlighted that these challenges render the need to include controversial societal issues in mathematics classrooms more systematically, in order to make future citizens familiar with such issues and the inherent mathematics in them. Ref. [2] described that including societal problems requires competencies from many scientific fields. It can be challenging for teachers to combat subject-based approaches. However, real-world problems are interdisciplinary and challenging; still, as citizens, we need to find ways to deal with them.

## 3. Theoretical Perspectives

Socio-critical modelling perspectives focus on exploring societal problems with mathematics as a critical tool [7]. Ref. [11] highlighted that socio-critical modelling perspectives serve dual goals, one within mathematics (e.g., how mathematical models support decision making), while the other is within societal spaces (e.g., addressing inequality and injustice). They emphasised that when combining modelling and societal issues, one should connect each of the components of the modelling process to societal issues (see Figure 1), rather than only some components in isolation.

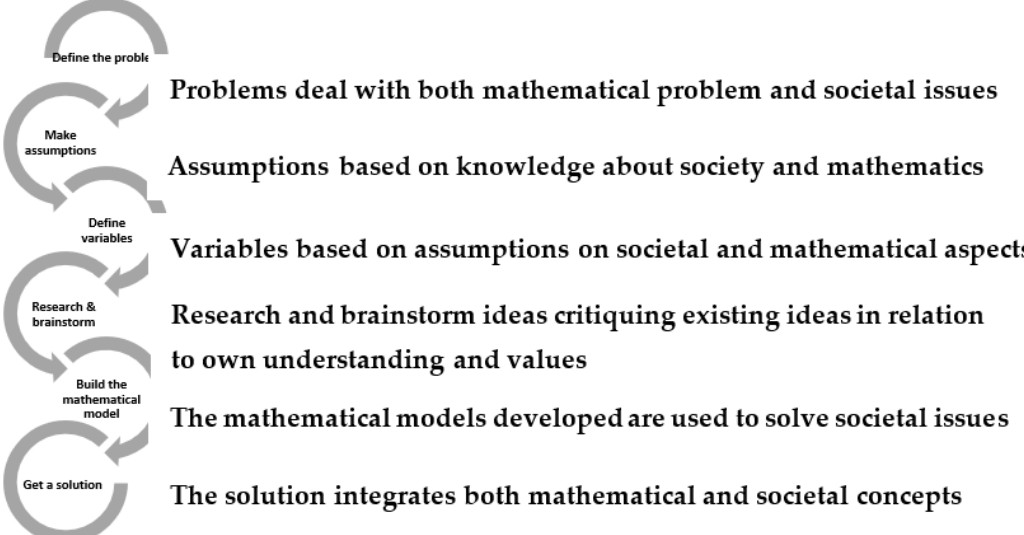

**Figure 1.** Combining mathematical modelling with societal issues [11].

Figure 1 illustrates the modelling process (on the left) and the process of combining mathematics and societal issues (on the right). Ref. [11] departed from the modelling process described by [44]. The modelling process is sometimes explained as different steps where students perform different activities, e.g., defining, structuring, and simplifying problems [16]. Ref. [11] suggested that all components in the modelling process should, ideally, include mathematical and societal considerations. That means it becomes insufficient to ensure that the problem deals with societal issues if the other steps of the process are not based on such factors as societal knowledge and understanding of mathematical data, or if the model developed does not aim to solve societal issues. Mathematical modelling and social justice, thus, become inherently intertwined through students' modelling processes. Integrating this duality in all steps of students' modelling processes is, thus, an aim when combining mathematical modelling and societal issues.

### 3.1. Modelling to Raise Awareness and Responsibility

Refs. [45,46] argued that mathematics education researchers, educators, and teachers have a responsibility to engage in societal issues. Mathematical modelling can be one way of assuming this. For instance, Ref. [47] highlighted that mathematical modelling could be used to raise awareness of critical issues in society. When investigating how socio-critical perspectives were present in a modelling project conducted by PTs in Argentina, one of the groups of PTs stated that their aim was "modelling to raise awareness" of their students rather than modelling "to obtain a super formula" (p. 573). The PTs focused on trash and recycling, and posed questions on the quantity and classification of trash. Mathematics became subordinate to the social aim, and mathematical modelling became a tool for understanding and reflecting on a phenomenon concerning the world outside the classroom. When introducing the modelling activity to the PTs, the teacher educators and researchers discussed that modelling activities should involve free choices of a real-world theme, interdisciplinarity, avoid pre-determined mathematical content, and offer reflection about mathematics, the model, and the societal role of mathematics.

Ref. [43] described an extended modelling cycle considering ethical, social, cultural, and economic aspects when controversial issues are in play. They emphasised that, to develop active and responsible citizens, mathematics education could include modelling activities where students could search for information about societal issues and contradicting discourses about scientific results or various reasonings grounded in ethical, social, or cultural considerations.

*3.2. Modelling to Empower Students and Take Actions*

Ref. [5] explained that mathematical modelling empowers citizens. It could provide them with the tools, the rights, and the responsibility to investigate critically and, if needed, reject mathematical arguments. They further highlighted that mathematical modelling could enable students to judge applications of mathematics used to describe and analyse aspects of our society. Refs. [5,6] described that the modelling activity with Barbie, which we mentioned earlier, is an example of using modelling as a critical tool for analysis. In a sense, it becomes a method of modelling for life and modelling with a purpose. Similar ideas are seen by [2,43], who argued that mathematical modelling is key to empowering students as responsible and active citizens. They problematised that mathematics education traditionally focused on concepts and competencies detached from societal implications. Therefore, they suggested that, alongside the teaching and learning of mathematical modelling, there are potential opportunities for developing understandings about socio–scientific issues, for inquiry-based learning, as well as numeracy, critical thinking, and 21st-century skills. When considering the professional development of teachers to support them realise these potential opportunities, modelling tasks involving controversial issues demanding ethical, moral and social reasoning and decision making could be discussed.

Ref. [9] visualised the modelling cycle next to a social justice cycle. In the latter cycle, it is emphasised to consider the broad social issue and build civic awareness and a sense of action. The modelling cycle is essential in modelling activities with students. If civic awareness and taking action are explicitly included in the modelling cycle, this can bring attention to the fact that these topics are relevant to consider when modelling. They described that, after the teachers in their study engaged in one example of a socio-critical modelling task, they began to imagine other relevant topics for modelling in their classroom, including immigration issues, recycling, and food insecurities.

Ref. [19] highlighted that teaching mathematics in critical ways is not an option in today's society, referring to current crises such as climate change and refugee situations. He stated that we, as teachers, have a responsibility towards future generations and to our planet, and what we do in the classroom matters. He used the phrase "reading and writing the world with mathematics" (p. 133) to describe a situation where students learn to use mathematics to investigate their society, understand forms of injustice, and enable them to act to change them accordingly.

Ref. [48] also emphasised action and stated that students could take action when critically investigating and reflecting upon real-world modelling problems. They described how dimensions of socio-critical modelling attempt to address multiple ways of working with real life, where students are supported to understand, explain, deal with, and suggest solutions to various problems. To take action on inequality issues, ref. [48] highlighted that students should be critically aware of mathematics as part of power structures in society.

Ref. [49], among other relevant studies, discussed that culturally relevant pedagogy can be strengthened through mathematical modelling, because students' backgrounds, knowledge, and experiences can be acknowledged to bridge home cultures and school. This could support students in developing critical consciousness, which refers to understanding societal systems and acting on them with a sense of agency [50]. Cultural issues are embedded in societal issues. For teacher education, they suggested that a focus on PTs posing open-ended modelling problems, making assumptions, and engaging in discussions and readings about culturally relevant pedagogy that could support understandings about the relations between societal and cultural issues in mathematics education.

## 4. Context of the Study

In our case study, we aimed to understand how an MTE integrated societal issues in mathematical modelling activities. The study took place in a teacher education institution in Norway as part of a larger design-based research project, "Learning about teaching argumentation for critical mathematics education" (LATACME). The project investigated what promotes or hinders PTs' learning about teaching argumentation for critical mathematics education in grades 1–7 multilingual classrooms. The MTE, the main subject of this case study, is the second author of this paper (Georgia) who was in her second year of working as a teacher educator when the study took place.

We investigated a course in the first semester of the second year of a five-year teacher education program, which included a period of school practicum. The course also included an obligatory assignment where PTs needed to describe and analyse their experiences of designing and implementing mathematical modelling in their practicum. The workshops were conducted before the practicum and lasted three hours each. Two out of three workshops of the first cycle and all three workshops of the second cycle took place digitally due to COVID-19 regulations. During the workshops, the main language spoken by the MTE was English, as her native language is not Norwegian. However, the MTE and PTs drew on Norwegian at times.

### 4.1. Data Collection

The data come from Georgia's two cycles of planning and implementing three workshops of mathematical modelling to three groups of PTs each year, as part of the mandatory mathematics education course. The empirical data included in this case study consist of the PowerPoint slides used in Georgia's workshops, her reflection notes before and after each workshop, transcripts from the audio-recorded workshops, and the PTs' written responses to tasks during the workshops, where available. Some sequences of the workshops were not audio-recorded because they were digital, and not all PTs had consented. The workshops aimed to introduce PTs to planning and implementing mathematical modelling activities in multilingual classrooms in grades 1–7. The theoretical framing of modelling in the workshops was based on socio-critical perspectives of modelling [7], where a modelling activity is understood as a problem (not an exercise) for the students, extracted from everyday life or sciences that are not pure mathematics (p. 294). Emphasis was given to problem-posing, framed within the ideas of activism and awareness of injustices in society [50,51]. To illustrate different perspectives of modelling activities and posing questions, Georgia used various examples from real-life situations, such as oil spills in the ocean and pollution caused by the accumulation of cigarette butts.

To support PTs in modelling activities, the group of MTEs in the teacher education program decided to use mathematics in three acts [52,53] as a teaching arrangement, or the adjusted approach; modelling in three acts [34,54]. Mathematics in three acts includes: identifying a problematic situation based on visual illustrations (e.g., pictures, films, graphs, and concretes) (Act 1), working in small groups to retrieve the necessary information to approach the problematic situation (Act 2), and solving and presenting solutions (Act 3). During Georgia's workshops, more emphasis was given to Act 1, connected to problem-posing. For example, she requested PTs to reflect on their role in supporting students to pose their own questions in a modelling activity, handling students' mathematical and non-mathematical questions, and taking action to change the situation identified as problematic.

As a beginning MTE, Georgia had been engaged in investigating her own teaching practice through action research [55,56]. However, her focus on investigating and improving her practice had been on integrating issues of language diversity rather than socio-critical modelling activities. The first author of this paper, Lisa, was also involved as an MTE during the LATACME project. Since we, as MTEs, shared similar genuine interests in mathematics education but came from different backgrounds and teaching experiences, we collaborated closely and decided to learn from each other's practices. For example, Lisa taught one

workshop on mathematical modelling to one group of PTs before Georgia. Before and after our respective workshops, we reflected on our plans as MTEs and discussed the needs we perceived that PTs had. In a prior study we conducted based on a related dataset within the LATACME project [34], we became aware of the impact that our decisions and examples can have on PTs integrating societal issues into mathematical modelling. Even though the focus of that prior study was on PTs' practices after Georgia's modelling workshop, in the present study we extend our understanding of the topic by looking further into the same MTE's practices.

*4.2. Data Analysis*

To understand how the integration of societal issues in modelling activities was exemplified in the MTE's practice, we first categorised the data into broad themes based on the content of the workshops and the reflection notes. The emergent themes were: pedagogical approaches, choice of literature, curricula, modelling tasks and activities, challenges of modelling, modelling competencies and sub-competencies, problem posing, modelling cycle, teaching in language-diverse classrooms, and socio-critical modelling perspectives. Then, we identified similarities and differences across the themes based on the theoretical framework discussed in the previous section. We analysed the data according to two themes inspired by components of the modelling cycle, particularly posing problems involving both mathematical and societal aspects [11]. Other analytical lenses were: modelling to raise awareness and responsibility [43,47] and modelling to empower and take action [5,6,18,49,50].

Therefore, we grouped the MTE's ways of exemplifying connections between modelling and societal issues into the following foci: problem posing, raising awareness, responsibility, empowering, and taking action. In the next section, we analyse and discuss examples of how these foci emerged in the MTE's practice. The research study follows the ethical guidelines from [57].

## 5. Findings and Discussion

We structure the discussion of the data in two parts: (1) raising awareness and responsibility by posing problems focusing on societal issues; and (2) modelling to empower and take action. In the first part, we describe three modelling activities where the combination of mathematical and societal aspects was exemplified in relation to problem-posing processes. In the second part, we describe an example related to the modelling cycle. We chose this example because it provides opportunities to understand how MTEs can connect the theoretical framing of mathematical modelling and socio-critical perspectives, such as taking action. The examples we discuss are taken as potentialities in order to gain insights about MTEs' practices and decision making rather than as "ideal" examples of approaching the combination of mathematical modelling activities and societal issues.

*5.1. Raising Awareness, Responsibility, and Problem Posing*

We identified three examples of modelling activities related to problem posing that Georgia drew on in different workshops: littering, Barbie, and oil spill.

The littering activity was related to PTs' hiking experiences, a popular everyday activity in Norway. Therefore, this context gave the MTE the possibility of combining reflections on a micro-level (PTs and students can observe littering in their daily lives) and a macro-level (littering is a nation- and global-wide issue), as indicated by [10]. This example was introduced to PTs as a potential modelling activity in three acts [52,53], named "Don't let your hiking go to waste" and developed as part of LATACME. This activity was designed during the LATACME project, see https://prosjekt.hvl.no/latacme/wp-content/uploads/2020/09/Modelling-tasks-hiking-ice.pdf (accessed on 13 July 2023).

In Act 1, Georgia showed a slide with authentic pictures of a local hiking area where littering was observed (Figure 2). The pictures included different kinds of trash, such as

plastic bags, glass bottles, cigarette butts, and cigarette packaging, between plants and in the lake by the hiking area.

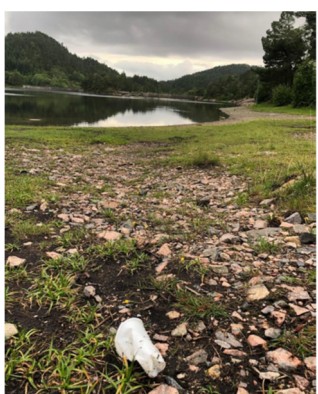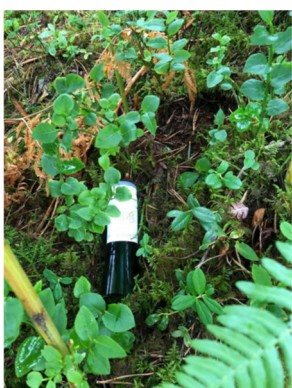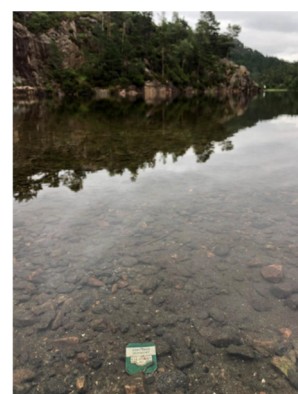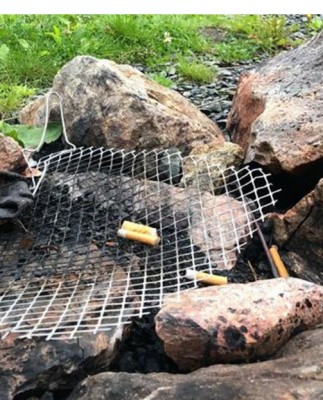

**Figure 2.** Four of the pictures used to bring awareness about littering. Photo from a hiking trail in Kanadaskogen, Bergen, in June 2020. Pictures taken by Camilla Meidell and printed with permission.

She asked PTs to discuss what they noticed in these pictures, identify problematic situations, and pose questions in small groups in a shared document. Some of the questions that PTs shared were: "Why is it important not to litter in nature?" and "Which distance should there be between each bin for people to throw waste in, instead of in nature?" One group of PTs wrote:

> *Problem solving: How to reduce litter in nature? Reasoning and argumentation: Why is it important that people do not throw rubbish in nature? How much trash do we find? How many trash cans? Where are the trash cans? Near places to eat? If we increase the number of rubbish bins, will the amount of rubbish in nature decrease? Statistics: How much plastic, paper, residual waste, and food waste do we find in nature? Make forms and tables.*

Problem posing can be challenging for students [12,16] and formulating questions in groups can be an opportunity to test out various questions they want to investigate. Engaging the PTs as learners of problem posing combined with being teachers preparing lessons for their students, in line with the ideas of [27], is one way of providing the PTs with the experience of posing questions when modelling. In this sense, PTs experienced problem posing around socio–environmental issues as learners, not only from their own perspectives but also from observing the problems their peers posed, similar to the PTs in the study of [29]. Later, PTs were requested to reflect as teachers and discuss how they could deal with possible challenges of problem posing processes in the classroom, such as dealing with the nature of students' questions, either mathematical ones or other questions that are (initially) non-mathematical (see [34] for related discussions). A focus on identifying mathematics can support students in seeing the mathematical aspects of real-world problems while bringing awareness of littering issues in society, similar to students who identified mathematics related to the refugee crisis in the study of [34].

Following the PTs' discussions and sharing of questions, Georgia included the following slide (see Figure 3):

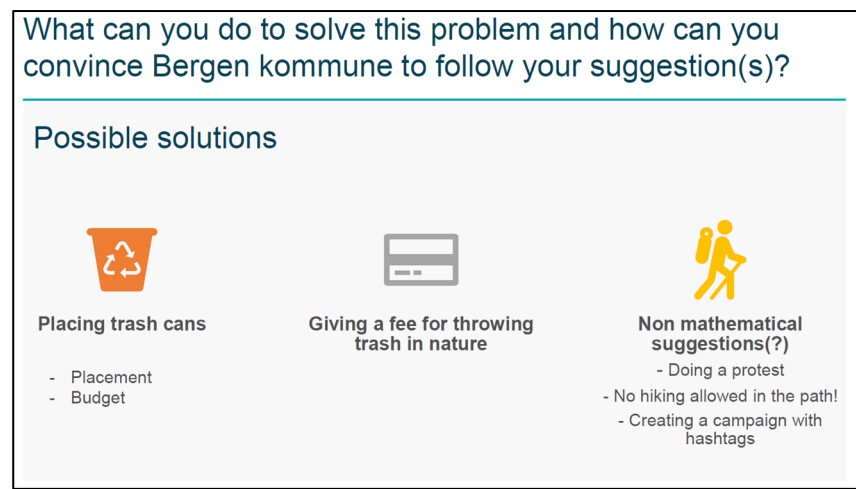

**Figure 3.** The PowerPoint slide from Georgia.

Georgia linked the littering activity with the idea of taking action by returning to PTs as learners and asking them to consider what they could do to "solve" the problem they had identified around littering and how they could convince the municipality where the hiking area belongs to follow their suggestions. Thus, the littering activity allowed Georgia to introduce a real-world problem combining mathematics and societal issues, highlighted as essential by [11]. Using the municipality as an audience in the modelling activity [58] can be considered a technique to work on the activity from the point of view of the agents who hold responsibility for the problematic situation or need to assume responsibility for its resolution. The first part of the MTE's question ("What can you do to solve this problem?") addressed PTs' personal responsibility as citizens, in line with the ideas of [2,43]. It also pointed towards potential actions, i.e., what the PTs can do, which can be seen as exemplifying the use of mathematics to change a problematic situation, following the ideas of [18]. In the second part of the question ("[...] and how can you convince Bergen municipality to follow your suggestions?"), PTs were asked to imagine the municipality as a "client" needing a solution. The idea of involving a client was highlighted by [11] as a fruitful approach to making the modelling task authentic. In this case, Georgia used the example of Bergen municipality as a client who needs to be convinced to pay attention to the problematic situation and, thus, broadened the littering problem from a societal to a political problem. However, the MTE did not ask PTs to work further with modelling the activity and the problems they had posed and, therefore, did not proceed to actually send suggestions to the municipality. Therefore, an opportunity was missed to exemplify making authenticity real and not imaginary, or, in [33]'s words, to use mathematics as a tool for socio–political change in society. As described by [58], the idea of an audience can change how PTs use mathematics in order to convince the audience about the significance of the problem at hand.

Following the question of convincing the municipality, Georgia presented information about the hiking area and more authentic pictures as different alternatives of what dimensions of the problem one could focus on. These different alternatives, or examples to explore, included: a geographical dimension (map and bins placement), a statistical dimension (personal opinions of the locals), an economic dimension (budget), an environmental–political dimension (taking action to protect the nature), and an ethical dimension (legitimacy). The diversity of dimensions reflects the complexity of considerations needing to be made when working on socio-critical modelling activities. The various dimensions could also support PTs' understanding of "modelling to raise awareness" rather than modelling for developing a "super formula", in alignment with the study of [47].

The geographical dimension included a map of the hiking area and its scale, which could be used as a potential model for placing regular trash bins or cigarette butt bins.

Here, the issues of informing hikers about the placement of trash bins and prohibiting them from littering were brought up as possible solutions for reducing littering in the area. The economic dimension included some of the expenses that the municipality would have to make to place and maintain trash bins. For instance, Georgia exemplified questions in a PowerPoint slide such as:

> *How much do the cans cost? How big should they be? In what shape? Does it matter? How much money does the maintenance of the trash cans cost? How much does the pollution of not putting up trash cans cost? Picking up trash, environmental costs, wildlife, and ecosystem.*

This set of questions suggested how posing questions based on the economic dimension and the cost of placing trash cans can have extensions in environmental dimensions, such as air pollution. This seems to have allowed Georgia to compare the cost of taking action to place trash bins with the accumulated costs of *not* taking any action to do so (e.g., the impact on the environment, wildlife, and ecosystem of the hiking area). By bringing up this comparison between taking action and not taking action, PTs could develop an understanding of working on socio-critical issues and become aware that it involves a responsibility to make a decision that will have an impact on society, whether the decision leads to a change or not. Littered cigarette butts have environmental impacts, such as leaching toxic chemicals into the ground and water (e.g., lead and arsenic) and being a major plastic polluter as the filter consists mainly of plastic fibres. Georgia introduced the environmental–political dimension to PTs with a number of possible (initially) non-mathematical suggestions for the littering problem. These suggestions included organising actions, such as protests and online campaigns, or considering whether hiking in that area should no longer be permitted to protect the nature and ecosystem within the hiking path and the lake.

Littering is also an ethical issue and can lead to facing ethical dilemmas. For instance, do we have an ethical responsibility for other people littering? Also, littering cigarette butts is sometimes considered acceptable, which can make it challenging to act on the problem. In the workshop, Georgia included ethical dilemmas that could arise if it became illegal to litter in the hiking area. For instance, she posed questions such as:

> *If I am not going to place trash cans or trash bins, should I place cameras? Is that another solution? Is that ethical now? Should I be . . . Should I suggest to Bergen kommune (municipality in Norwegian) to do that? And who is going to be responsible for that? How much is going to be the fee for people who throw trash?* (workshop 1, p. 7)

These questions made available to PTs ethical concerns that may challenge everyday structures that are often taken for granted, such as the techniques used to identify individuals who litter, the settling of fees, and the municipality's role. Norwegian culture is often associated with being outdoors and protecting pristine nature from littering. While identifying people littering can be easily achieved by monitoring, people go to hiking areas to relax, so surveillance of citizens is very intrusive, and ethical considerations should be made.

In the modelling activity "Barbie in real life", Georgia showed a picture (picture: https://www.cbsnews.com/news/life-size-barbies-shocking-dimensions-photo-would-she-be-anorexic/ accessed on 13 July 2023) of a scaled-up Barbie next to a picture of a real woman, indicating that Barbies are often far from being realistic representations of women. Based on the pictures, a question was posed by the MTE, suggesting a potential modelling task: "What would Barbie look like if she was one of us?" Taking the starting point in Barbie and asking students to imagine how she would look in real life was also described by [5,6,39,41]. In the case of Georgia, according to her preparation notes, the choice of the Barbie context as an example of setting up socio-critical modelling activities was made in order to make links to PTs' obligatory assignment where the "Bungee Jump Barbie" task [59] was recommended by the team of MTEs. In the "Bungee Jump Barbie" task, students investigate how long the rope (rubber band) holding Barbie should be if she is dropped from a certain height to the



point of just touching the ground's water surface. Georgia compared the reality factor of the two Barbie activities:

> *The use of modelling as content, in cases like this bungee jumping Barbie, (...) where the goal of the bungee jumping of Barbie is to learn math, like measuring distance and finding patterns. ( ... ) That was a problem that was already set by a teacher. But from a more social and critical perspective, what would be interesting would be something like: Is this Barbie model realistic? Is it real? Which could also involve very interesting discussions around proportional thinking, different analogies, functional thinking, symmetry, measuring, and modelling. ( ... ) My point here is that different modelling perspectives ask different questions, so not all questions start from a situation asking how much or how many there are* (workshop 2, pp. 10–11).

In this excerpt, Georgia referred to the "Barbie in real life" activity as an opportunity to extend the "Bungee Jump Barbie" and combine mathematics with socio-critical issues related to body image. Discussing issues of body image or eating disorders can be challenging for students and teachers because it can affect students in the classroom at a personal level. It can, therefore, be difficult for PTs to combine and balance the mathematical and societal aspects of this modelling problem. Georgia suggested various mathematical topics that could be relevant when exploring the "Barbie in real life" modelling task, similar to the task described by [16,37,38]. Thus, the "Barbie in real life" modelling activity could engage students in such as proportional reasoning, facing similar mathematical challenges described in their research. However, the PTs may find it challenging to recognise such problems as mathematical. For instance, while Georgia introduced various societal issues, the following dialogue took place during the first workshop (pp. 6–7):

> *PT: Could you please explain how you're playing this (referring to the societal context) into math or the modelling? How it comes together?*
>
> *Georgia: What is math for you? (pause) That does not look like math?*
>
> *PT: Doesn't really . . . (pause) But if it's like you're taking focus on argumentation and reasoning, and then quite (inaudible) to mathematics . . . then I'm in, then I can understand.*

The PT asked Georgia to connect the societal issues to mathematics explicitly. Instead of explaining, Georgia turned the question around and asked the PT what she considered as mathematics. The PT started pondering before she related the question to argumentation and reasoning, which are part of the mathematical competencies in the Norwegian mathematics curriculum [14]. "Barbie in real life" could encourage the students to investigate and engage in societal issues, such as body image stereotypes, societal expectations, and eating disorders. Bringing awareness of the matters of eating disorders could involve knowledge about the illness itself but also to socio–political dimensions of this illness. Students could investigate the social and economic cost of how this illness affects society [60] or how insurance companies treat this disorder and how this affects marginalised groups in society [61]. To extend a "feminist focus", as reported by some of the PTs in the study of [41], one could also focus on the occurrence of eating disorders in males, a historically under-researched area (males compromise about one in four cases in bulimia nervosa and anorexia nervosa [62].

The third modelling activity was the "Oil Spill". Georgia started by showing a map representing the span of an oil spill in the Gulf of Mexico in 2010 (Figure 4). This oil spill is historically one of the largest marine accidents, with more than 200 million gallons of oil being spilt [63]. Four questions accompanied the map (see Figure 4).

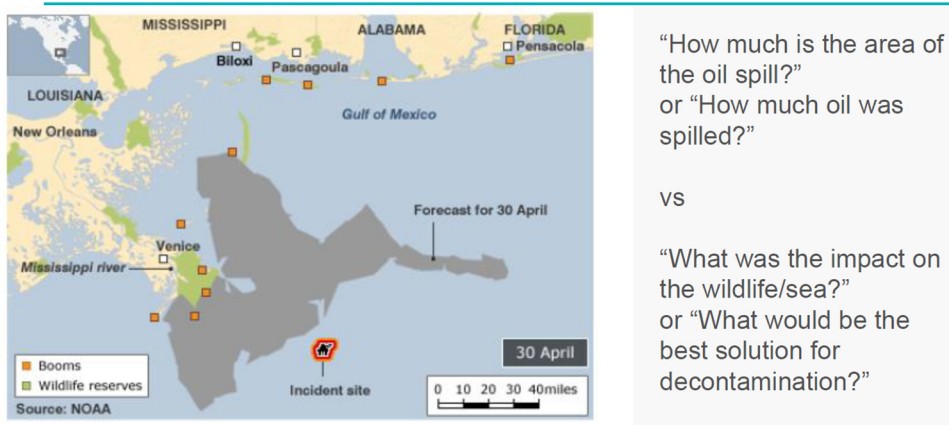

**Figure 4.** The slide in the fourth, fifth, and sixth workshops depicting a map of the oil spill in the Gulf of Mexico and the four questions. Map from "Oil spill in Gulf of Mexico in maps and graphics" by BBC (NOAA) [64] (http://news.bbc.co.uk/2/hi/americas/8651333.stm, accessed on 11 July 2023).

Each question in Figure 4 exemplifies posing problems for potential modelling activities. The first two questions suggest a mathematical focus on quantifying and estimating. Estimating the area or the amount of spilt oil is not a straightforward task. It involves (mathematical) challenges, such as deciding the surface area and the average thickness or risk analysis. It also could include political implications, such as whether oil drilling is still worth it and who should bear the costs. It resembles the modelling task described by [36], where students investigated the Great Pacific garbage patch and posed questions involving the extent, environmental effects, and potential actions. Estimations of geographical areas or crowds from a socio–political perspective are described by researchers such as [65–67]. While [65] discussed various estimations on casualties in the war in Iraq, the two latter researchers investigated how students estimated attendees in protests in Chile and Greece, respectively. Combining mathematical estimations with the political implications of oil spills can be fruitful in developing an integrated understanding that mathematics is not just a tool separated from decisions in society. Making PTs familiar with such decision- making through modelling tasks has the potential to develop responsible citizenship, according to [3].

In her reflection notes, Georgia wrote about the "Oil Spill" modelling activity:

*I think it's interesting and critical. ( . . . ) It is again a modelling task with Critical Mathematics Education (CME) concerns, and it is much more realistic than the spilled ink task, even if they share the same (mathematical) ideas. One option could be to show both tasks and ask about (PTs') opinion and compare the tasks since they are similar and have similar (mathematical) ideas and aim* (workshop 4, p. 4).

She referred to another mathematical task she had used earlier in her teaching, where students/PTs were expected to estimate the area of ink spilt into a paper surface. Similar to comparing the "Barbie in real life" with the "Bungee Jump Barbie", the comparisons made between the "spilt ink" task and the "Oil Spill" activity allow the MTE to exemplify reflections on modelling activities that extend mathematical skills to knowledge around socio-critical issues, in this case, environmental phenomena. Although the "Oil Spill" modelling activity occurs in the Gulf of Mexico, it is transferable to other places worldwide, as well as in the context of Norway. In Norway, where the PTs live, a potential oil spill could have catastrophic impacts on the wildlife in areas such as the Lofoten islands and the Arctic Ocean. To prevent this from happening, the Norwegian government has temporarily suspended all oil exploration of Lofoten. As a nation profiting from oil and gas exploitation and having a long coastal line at risk of being affected by oil spills, it is

imperative to consider risks and impact assessments when deciding whether to drill or not. However, estimating and calculating the potential damage to wildlife can be challenging and economic considerations are often prioritised. Therefore, introducing questions with a mathematical and with socio–environmental focus could exemplify to PTs that a modelling activity could include considering more quantifiable dimensions (e.g., income from oil and gas) and less quantifiable dimensions (e.g., value of wildlife and pristine nature). This example could provide opportunities for PTs to support their students to reflect on what matters in a model, what kind of assumptions should be included, and how to deal with assumptions not included.

In Norway, the industry is part of funding their welfare state and is part of the Norwegian culture [68]. Integrating cultural knowledge into mathematical problem solving is in line with the ideas from [49]. The students' cultural knowledge about oil is not a problem selected for a particular group of students but relates to everyone living in Norway. It concerns those working in the industry (or who have parents working) and those benefitting from that industry (e.g., through free health services and school). However, although the petroleum industry has various positive impacts on Norwegian society, it is crucial to investigate problematic issues of the industry critically. Living in an oil-rich nation should bring about responsibility, as described by [2,69], and awareness of the potential consequences of accidents or the long-term impacts of climate change. Students could develop their competencies to understand and act on societal systems as well as their critical consciousness [50]. There are ongoing public and political debates in Norway, such as stopping to search for more oil and gas and preventing explorations in certain areas (such as Lofoten Islands and the Arctic).

### 5.2. Modelling to Empower and Take Actions

Georgia included theoretical dimensions of modelling as described by [16,38]. She used several visualisations of this process when introducing the modelling cycle. In one of the workshops, she added the word "Action!!" in the cycle described by [70] (see Figure 5).

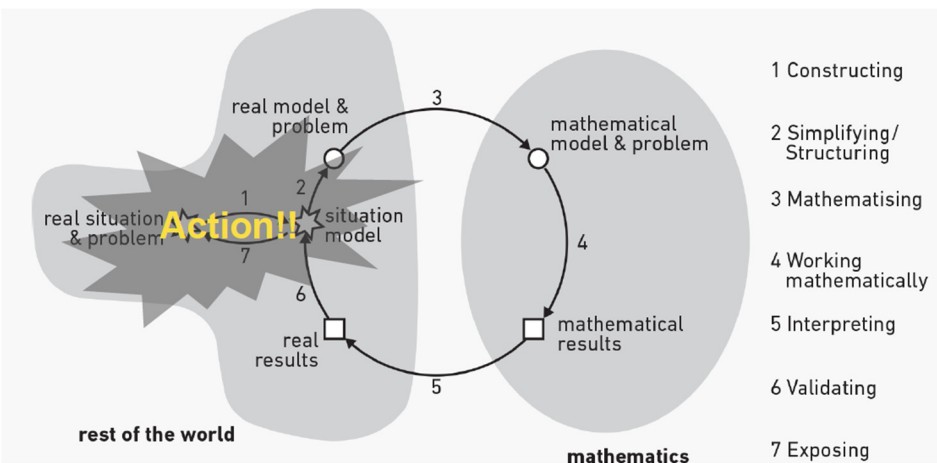

**Figure 5.** The slide in the second workshop showing the word "Action!!" added in the modelling cycle by [70].

While showing the modelling cycle in Figure 5, she said:

*But ( . . . ) after working with a critical perspective, what I'm always interested in as a problem solver, and a modeller, is doing something in the end that can change something in a situation that is really problematic. I have worked with all these numbers and problems that concern society, the community, the school, and my home. I have a responsibility as an individual and with others that I have worked with also. To take action, to do something with the power of math, the numbers, the power of the math that math has given me.*

> *Because that is another aspect of math, it empowers the problem-solver* (workshop 2, page 12).

She started by emphasising a critical mathematics perspective connected to her own role as a problem solver and modeller in changing a societal problem. She continued by connecting her responsibility as an individual with taking action with the help of mathematics, ideas which have been introduced by [19,50]. By forwarding these ideas to PTs, Georgia exemplified taking a form of action. Further, she took an opportunity to exemplify that mathematics is not neutral but has the potential to empower, in line with what [2,5,43] described. When Georgia added and highlighted the word "Action!!" into the modelling cycle, as shown in Figure 5, she extended the modelling cycle to include socio-critical perspectives. Similar extensions have been described by researchers such as [9,43,48], who have all presented various versions of the modelling cycle to combine social-critical aspects with the traditional cycle. Based on these studies, as well as on [11], who emphasised that socio-critical perspectives should be integrated in all parts of the modelling process, Georgia could have used this opportunity to be more explicit in exemplifying that the need for "Action!!" concerns all parts of the cycle, rather than just the end.

Later, the MTE further elaborated on how this responsibility and action could manifest within aspects of one's everyday life:

> *For example, if I work with a problem that is about plastic, as in plastic packaging, trash in any way, or cigarette butts, what will I do in the end? Will I actually tell my parents and my family when I see someone throwing plastic packaging on the ground or cigarette butts? How will I convince them that this is not rational behaviour, not responsible, because I have done the math? ( ... ) That is why I am adding this action in this part of the modelling cycle. It is important that you take action. When you come to work with a problem, it is a social problem, and I have some responsibility. I have "ansvar" ("responsibility", in Norwegian) to tell others and make my voice heard* (workshop 2, pp. 12–13).

With indirect questions to PTs, she brought examples from everyday life, plastic packaging, trash, and cigarette butts, and connected them to the importance of taking action and assuming responsibility. By doing so, the MTE could exemplify that action is not a single, limited, and finite doing but that working on a modelling activity with socio-critical concerns comes with a commitment to action. For example, it involves a kind of societal commitment to being a responsible citizen and spreading awareness about the problem. Dealing with societal issues such as trash can include a focus on systemic and global challenges, e.g., as described by [36] in the garbage patch activity or by [47] in the trash and recyclable collection in Cordoba city. It can address systemic injustice, as described by [9] or [33], since those dealing with consequences of littering may not be the ones causing it. However, it can also have an individual focus. Reflecting on our behaviour can bring awareness of what we can do as individuals, which is important to consider for taking action. For example, when referring to dealing with other people littering, the MTE connected the commitment to action with a personal dilemma one might face in everyday life. That is because, on the one hand, it can be uncomfortable to steer others' behaviour and be perceived as moral policing, but, on the other hand, having "done the math" leaves one having to make decisions based on what is mathematically right and ethically appropriate. Therefore, according to Georgia, it could still be challenging to "convince" people to behave rationally.

Encouraging the ideas of responsibility is a critical part of mathematics teachers' and teacher educators' roles, according to several studies, such as those of [45,46]. Yet we can start questioning how much of that responsibility is shared and should be "transferred" to students. Students are often innocent bystanders, and it is unfair to burden them with the responsibility of socio-critical issues, such as climate change. However, many students are already aware of these problematic issues, and having spaces to make their voices heard and take action has implications both within and outside the classroom borders.

## 6. Conclusions

In this article, we discussed integrating societal issues in mathematical modelling in teacher education. Based on the case of one MTE, we identified ways by which this integration was exemplified and the potentialities for PTs' and students' learning.

For example, we found that the MTE included three modelling activities, "Littering", "Barbie in real life", and "Oil Spill", with a focus on problem posing. Focusing on posing problems and identifying mathematics can support PTs and students in recognising the mathematical aspects of real-world issues while simultaneously bringing awareness of problematic issues in society. The MTE's choice of these contexts was also an implicit exemplification because they were relevant to PTs' everyday life (hiking, oil spill) and their coursework (Barbie). As well, adding the municipality as an audience in the modelling activity of "Littering" as an imagined client [11,58] was also a practice that could contribute to PTs' understandings about the purpose and implications of working on a modelling activity that concerns society. Another practice of the MTE was bringing in various problem dimensions, such as geographical, statistical, economic, and environmental–political. Showing a multiplicity of dimensions within the same modelling activity, thus, could support PTs in understanding the complexity of such activities and the different alternatives that are available. For instance, in the economic dimension of the littering activity, Georgia compared two alternatives where critical choices were required: that of placing and not placing trash bins.

Further, we identified that Georgia compared the "Barbie in real life" and the "Oil Spill" activities to tasks not focusing on socio-critical perspectives [5,6,39,41]. Considering that PTs were at the beginning of their second year of teacher education, showing such comparisons could support them in designing socio-critical modelling activities by challenging and broadening the tasks they are more likely to have encountered as mathematics learners and teachers. Knowing that many PTs have difficulties with developing socio-critical modelling activities, starting from challenging what they can already access might seem less overwhelming for them than dealing with the uncertainty of constantly developing new activities.

Lastly, the MTE focused on taking action and assuming responsibility [19,50]. This was connected to: theoretical perspectives of teaching and learning modelling, such as when adding the word "Action!!" in the modelling cycle, and to self-reflections on being committed to taking action and being a responsible citizen empowered by mathematics [2,5,43].

Even though we identified multiple situations where the MTE combined mathematics and socio-critical perspectives, there is still room for improvement both in the individual work of MTEs and within the teacher education community. Limitations of this case study include risks of bias. Also, the study took place during COVID-19 restrictions, and although the PTs interacted in online groups, they were not physically in the same room, and their communication was influenced by the boundaries of digital tools (e.g., some PTs did not have cameras).

Implications of this study are that socio-critical issues should not be separated from mathematics education courses about modelling; rather, the focus should be given to both mathematics and societal issues in all parts of the modelling process [11], if PTs and students are to be empowered as responsible citizens. Further research could investigate more holistically the modelling process when combining mathematical aspects and societal issues. Future research could also concern the support MTEs have when including mathematical modelling and societal issues. In our case study, the MTE (Georgia) was supported by her Ph.D. supervisors, both experienced researchers in research fields such as critical mathematics education, and some of her colleagues, including the first author (Lisa). Thus, a small informal community supported the MTE in discussing and reflecting on her teaching. However, as [33] highlighted, combining socio–political issues with mathematics can be challenging and a well-established community of MTEs engaged in socio-critical issues of modelling activities is needed. Future research could explore how MTEs can be

supported to include societal issues, particularly novel teacher educators or educators who have not previously included socio–political issues in their mathematics teaching.

**Author Contributions:** This paper is based on research by G.K. during her Ph.D. Georgia collected, organised, and transcribed all the data. L.S. wrote the draft preparation. Both authors have contributed equally during analysing data and writing the paper. All authors have read and agreed to the published version of the manuscript.

**Funding:** This research was funded by the Research Council of Norway (grant number 273404) through the project "Learning about teaching argumentation for critical mathematics education in multilingual classrooms" (LATACME) at the Western Norway University of Applied Sciences.

**Institutional Review Board Statement:** The study was conducted in accordance with the Declaration of Helsinki, and approved by the Institutional Review Board of Sikt Norwegian Agency for Shared Services in Education and Research (protocol code 950095 and from 1 September 2018 until 31 December 2023).

**Informed Consent Statement:** Informed consent was obtained from all subjects involved in the study.

**Data Availability Statement:** Due to privacy restrictions and ethical reasons, the dataset connected to this study cannot be made public.

**Acknowledgments:** We acknowledge Camilla Meidell, former colleague, for her contributions in designing the activity "Don't let your hiking go to waste".

**Conflicts of Interest:** The authors declare no conflict of interest. The funders had no role in the design of the study; in the collection, analyses, or interpretation of data; in the writing of the manuscript; or in the decision to publish the results.

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
