# Peer review of "Integrating Societal Issues with Mathematical Modelling in Pre-Service Teacher Education"

_education, doi:10.3390/educsci13070721_

Round 1

Reviewer 1 Report

The article presents a very pertinent research focused on a relevant issue: how can the pre-service teacher education adress the integration of societal issues with mathematical modelling? I thoroughly appreciated it namely the deep way as the several theoretical perspectives were presented and discussed. 

It highlighted social-justice oriented issues connecting them to the modelling process. But the social issues are not restricted to inequality and injustice. Since the three examples presented in the results are not related to social-justice issues, I think that this emphasis is not adequate. 

The article does not present transcripts from the audio-recorded workshops nor  the PTs’ written responses to tasks during the workshops, despite they are referred as empirical data in Methods section. 

Act 1 focused in the article includes: identifying  a problematic situation based on visual illustrations (e.g., pictures, films, graphs, concretes). Therefore, only int the first example of Littering there is evidence of PT’s questions. In the other two examples, the questions are posed by the MTE, which offer a partial view of the activity developed during the workshops. It could enrich the article if the problematic situations identified by the PTs were presented. The focus on problem posing is limited to problem posing made by the MTE.

Author Response

It highlighted social-justice oriented issues connecting them to the modelling process. But the social issues are not restricted to inequality and injustice. Since the three examples presented in the results are not related to social-justice issues, I think that this emphasis is not adequate. 

We have included a paragraph in the literature review, where we elaborate on how we understand societal issues and social justice in this paper and how this relates to (critical) mathematics education, and the three examples.

The article does not present transcripts from the audio-recorded workshops nor  the PTs’ written responses to tasks during the workshops, despite they are referred as empirical data in Methods section. Act 1 focused in the article includes: identifying  a problematic situation based on visual illustrations (e.g., pictures, films, graphs, concretes). Therefore, only int the first example of Littering there is evidence of PT’s questions. In the other two examples, the questions are posed by the MTE, which offer a partial view of the activity developed during the workshops. It could enrich the article if the problematic situations identified by the PTs were presented. The focus on problem posing is limited to problem posing made by the MTE.

We have added more quotes from the participants (both teacher educator and pre-service teachers) and included other empirical data (pictures and images of PowerPoint slides) to make it more vivid and relate more explicitly to the data collection methods. This was done in the activity Littering, and in the activity Barbie. We have included both oral and written responses. In the Oil spill activity, we do not have audio-recording from the PTs (mainly because this was done digitially due to COVID19-restrictions), and this is now elaborated more about in the methodology section.

Reviewer 2 Report

This is a worthy study and one that i think needs to be explored more deeply not only in mathematics but in all curricular fields! Here are a few suggestions:

1. Add MTE with definition in key words

2. Since this is a Case Study Design (that I first encountered in line 65), I think that this should be mentioned in the Abstract. You only explain "the case" beginning in line 306 

3. Present a clearer definition of mathematical modelling in the introduction--your figures in the Results section does that!

4. Line 297 “we aim” -- instead use past tense

5. Figures in results section—great!!

Good--look for use of tenses.

Author Response

 Add MTE with definition in key words

Added mathematics teacher educator as a key word

Since this is a Case Study Design (that I first encountered in line 65), I think that this should be mentioned in the Abstract. You only explain "the case" beginning in line 306 

Included the word case study in the abstract

Present a clearer definition of mathematical modelling in the introduction--your figures in the Results section does that!

Provided an explanation of mathematical modelling in the introduction

Method: Line 297 “we aim” -- instead use past tense

Changed «we aim» to past tense

Also, looked through tenses in the document.